# Cross-modal Associations in Vision and Language Models: Revisiting the Bouba-Kiki Effect

**Tom Kouwenhoven**[*]**, Kiana Shahrasbi, Tessa Verhoef**[*]
Leiden Institute of Advanced Computer Science
Leiden University, The Netherlands
{t.kouwenhoven, k.shahrasbi, t.verhoef}@liacs.leidenuniv.nl

## Abstract

Recent advances in multimodal models have raised questions about whether vision-and-language models (VLMs) integrate cross-modal information in ways that reflect human cognition. One well-studied test case in this domain is the bouba-kiki effect, where humans reliably associate pseudowords like 'bouba' with round shapes and 'kiki' with jagged ones. Given the mixed evidence found in prior studies for this effect in VLMs, we present a comprehensive re-evaluation focused on two variants of CLIP, ResNet and Vision Transformer (ViT), given their centrality in many state-of-the-art VLMs. We apply two complementary methods closely modelled after human experiments: a prompt-based evaluation that uses probabilities as a measure of model preference, and we use Grad-CAM as a novel approach to interpret visual attention in shape-word matching tasks. Our findings show that these model variants do not consistently exhibit the bouba-kiki effect. While ResNet shows a preference for round shapes, overall performance across both model variants lacks the expected associations. Moreover, direct comparison with prior human data on the same task shows that the models' responses fall markedly short of the robust, modality-integrated behaviour characteristic of human cognition. These results contribute to the ongoing debate about the extent to which VLMs truly understand cross-modal concepts, highlighting limitations in their internal representations and alignment with human intuitions.

## 1 Introduction

Recent advances in multimodal models that integrate vision and language have brought artificial intelligence a step closer to understanding the world in ways that resemble human experience and cognition. These models, which learn from vast amounts of paired visual and textual data, have demonstrated impressive capabilities in tasks such as image captioning, visual question answering, and cross-modal retrieval (Radford et al., 2021; Li et al., 2022, 2023). Yet it remains unclear whether these models integrate visual and linguistic information in ways that parallel human cognitive processes. In this paper, we investigate whether VLMs exhibit human-like patterns of association between abstract visual shapes and unfamiliar words. Particularly, we focus on one of the most widely studied cross-modal test cases in human cognition, the bouba-kiki effect, in which people consistently associate pseudowords like 'bouba' with round shapes and those like 'kiki' with jagged shapes (Ramachandran and Hubbard, 2001; Maurer et al., 2006a; Ćwiek et al., 2022). These associations have recently become a test case for evaluating whether language models trained on large-scale data show similar patterns (Alper and Averbuch-Elor, 2023; Verhoef et al., 2024; Loakman et al., 2024; Iida and Funakura, 2024). Results, however, have been mixed. While Alper and Averbuch-Elor (2023) found overwhelming evidence for a bouba-kiki effect in CLIP and Stable Diffusion, other

---

[*]Equal contribution

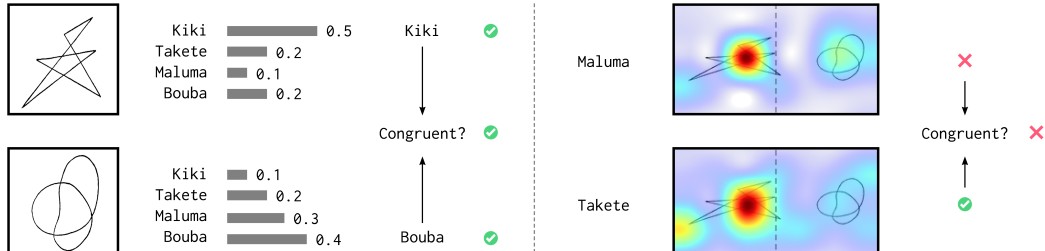

Figure 1: An overview of the two complementary methods used. On the left, we calculate the probabilities for each label across the four original pseudowords (note that the number of labels varies per label source) for each image shape and select the label with the highest probability (values are exemplary). On the right, we use concatenated image pairs and their labels as targets to calculate attention patterns with Grad-CAM and select the shape with the highest sum of attention.

studies have found less convincing patterns across different VLMs, testing methods and datasets (Verhoef et al., 2024; Loakman et al., 2024; Iida and Funakura, 2024).

This work therefore revisits this effect by thoroughly testing two versions of CLIP (Radford et al., 2021): ResNet (He et al., 2016) and ViT (Dosovitskiy et al., 2021). We focus on CLIP because, among four models (CLIP, BLIP2, ViLT, and GPT-4o) tested, Verhoef et al. (2024) found that CLIP demonstrated the most promising alignment with a human-like bouba-kiki effect. This aligns with previous work demonstrating that CLIP outperforms other models in capturing human-like decision patterns (Demircan et al., 2024). Perhaps most importantly, CLIP often acts as a foundational model in state-of-the-art VLMs (see section 3). If a 'base' model does not exhibit human-like associations, it is difficult to imagine that a model using that base model as a backbone will show human-like preferences without explicit fine-tuning on relevant cross-modal associations. Unravelling how these 'base' models represent cross-modal information additionally benefits our understanding of limitations in, for example, spatial reasoning (Thrush et al., 2022), (in-context) shape-colour biases in VLMs (Allen et al., 2025), and emergent communication setups (Kouwenhoven et al., 2024).

By probing these models' preferences for shape-word association in novel contexts, we aim to shed light on the nature of their internal representations and their alignment with human intuitions. When these differ, training models to develop human-like prior expectations or instilling them into models may help and could even improve learning efficiency (Lake et al., 2017). Doing so is essential, since a flexible, grounded understanding of this kind could enable conceptually fluent, natural human-machine interactions, in which machines intuitively understand what we mean, even in unfamiliar settings. To this end, we employ two complementary methodological approaches that address the same cross-modal associations from different perspectives, thereby minimising the limitations of relying on a single evaluative framework. Despite this comprehensive strategy, we do not find compelling evidence for the bouba-kiki effect. This work contributes in the following ways:

(1) We test for the bouba-kiki effect in VLMs in ways that are as close as possible to how it has been tested with humans in the past. This enables direct comparisons between our results and human findings, building on and expanding the work by Verhoef et al. (2024). Tested comprehensively, we find that models, compared to humans, do not make consistent cross-modal associations.

(2) We introduce a novel way to test models using Grad-CAM (Selvaraju et al., 2020), a method from the model interpretability literature, to look more closely at visual processing in the bouba-kiki context. Strengthening the robustness of our results, it reveals that the models do not explicitly focus on bouba-kiki related shape-specific features.

## 2   Related work

First, we discuss how prior work on cross-modal associations in human language and cognitive processing has shown that non-arbitrariness is both pervasive and affects how we learn, process and develop language. We then present previous studies that have tested for the bouba-kiki effect in VLMs.

## 2.1 Cross-modal associations in human language

Traditionally, it has been argued that mappings between words and their meanings are largely arbitrary. Hockett (1960) uses the words 'whale' and 'microorganism' as an example: 'whale' is a short word for a large animal, while 'microorganism' is the reverse. However, growing evidence from fields like cognitive science, language evolution and (sign language) linguistics suggests non-arbitrary form-meaning mappings are more widespread in human language than initially thought. This suggests that it should be considered a general property of human language (Perniss et al., 2010), shaped by cognitive mechanisms similar to those involved in the bouba-kiki effect. Especially when looking beyond Indo-European languages, 'iconic' mappings, where word forms seem to resemble their meanings, appear to play a significant role in many languages (Perniss et al., 2010; Dingemanse, 2012; Imai and Kita, 2014). Some languages have specific word classes where characteristics of the meaning are mimicked or iconically represented in the word. For example, Japanese ideophones allow speakers to depict sensory information through word forms, such as saying *nuru nuru* when describing something as 'slimy' and *fuwa fuwa* when it is 'fluffy' (Dingemanse et al., 2016). Similarly, in Siwu, *pimbilii* means 'small belly', while *pumbuluu* refers to 'fat belly', and non-speakers of those languages can typically guess the meanings of such words (Dingemanse et al., 2016). Even in languages not considered rich in sound symbolism, such as English and Spanish, vocabulary items from specific lexical categories, such as adjectives, are rated relatively high in iconicity (Perry et al., 2015).

Evidence for strong associations between speech sounds and particular meanings has been found in a broad sample of vocabularies from two-thirds of the world's languages Blasi et al. (2016). Iconic mappings are not only widespread in the world's languages, they also help both children Imai et al. (2008); Perry et al. (2015, 2018) and adults Nielsen and Rendall (2012) learn new words more easily. They, moreover, allow us to communicate successfully even when a shared language is absent or existing vocabulary is insufficient, because cross-modal associations lay the groundwork for the negotiation of novel words and their meanings (Ramachandran and Hubbard, 2001; Cuskley and Kirby, 2013; Imai and Kita, 2014). Shared cross-modal representations enable people to instantly interpret unfamiliar words even on first exposure by directly linking sensory experiences with meaning. Indeed, experiments with humans that studied the influence of cross-modal preferences on the emergence of novel vocabularies show that iconic strategies are frequently adopted when word forms and meanings can be intuitively mapped, and they help to communicate successfully (Verhoef et al., 2015, 2016b,a; Tamariz et al., 2018). An understanding of such mappings, however, requires human-like integration of multi-sensory information or an awareness of common cross-modal associations. While multimodal computational models have demonstrated remarkable capabilities across a range of tasks, the internal mechanisms through which these systems form and link representations remain opaque. In particular, it is still an open question whether these models integrate visual and linguistic information in ways that mirror human cognitive processes.

## 2.2 Bouba-kiki effect in VLMs

The bouba-kiki effect, as a specific example of visuo-linguistic processing, has been studied in VLMs before; however, the results so far seem to be conflicting. One key study in this space is the innovative work by Alper and Averbuch-Elor (2023), who convincingly demonstrated that patterns aligning with the bouba-kiki effect are reflected in the embedding spaces of CLIP and Stable Diffusion. They used Stable Diffusion to generate images based on pseudowords that were carefully designed to reflect phonetic properties associated with sharp or round shapes. Specifically, they used CLIP's text encoder to embed prompts containing either pseudowords or descriptive adjectives, while the images generated from pseudoword-based prompts were passed through CLIP's vision encoder. This setup allowed both text and image representations to be analysed within the same multimodal embedding space. The embedding similarity between the pseudowords and the adjective or image representations was then used to compute geometric and phonetic scores, indicating alignment with sharp or round associations. They concluded that their findings indicate strong evidence for the existence of cross-modal associations in VLMs. Typical explanations given for the presence of a bouba-kiki effect in humans include experience with acoustics and articulation (Ramachandran and Hubbard, 2001; Maurer et al., 2006b; Westbury, 2005a), affective–semantic properties of human and non-human vocal communication (Nielsen and Rendall, 2011), or physical properties relating to audiovisual regularities in the environment (Fort and Schwartz, 2022). This renders the conclusion by Alper and Averbuch-Elor (2023) somewhat surprising, as these explanations all involve situated, real-

world experience with a body and the environment, something these models entirely lack. Moreover, limitations of VLMs in visual grounding have been observed in many other domains (Thrush et al., 2022; Diwan et al., 2022; Kamath et al., 2023; Jabri et al., 2016; Goyal et al., 2017; Agrawal et al., 2018; Jones et al., 2024), suggesting that these models do not integrate textual and visual data in a manner that is human-like. Nonetheless, this work presented an innovative method for testing the bouba-kiki effect in VLMs. It convincingly demonstrated that VLMs encode relationships between word forms and semantic concepts related to roundness and jaggedness. As described previously, these associations are indeed abundantly present in human languages, and prior work with text-only language models has also shown that these models can detect such regularities (Abramova and Fernández, 2016; Pimentel et al., 2019; de Varda and Strapparava, 2022; Marklová et al., 2025).

Interestingly though, Iida and Funakura (2024) replicated Alper and Averbuch-Elor's experiments for Japanese and found that Japanese VLMs did not exhibit the expected bouba-kiki effect, even though Japanese is a language rich in sound-symbolism (Dingemanse, 2012), and speakers of Japanese display the strongest bouba-kiki effect compared to speakers of 25 other languages in a study by Ćwiek et al. (2022). These findings suggest that Alper and Averbuch-Elor's method may not capture true sensory mappings, but instead detects regularities between word forms and meanings that are language-specific rather than universal. For example, sounds like */p/*, which are typically linked to sharpness according to the bouba-kiki effect, are strongly associated with roundness in ideophonic Japanese words, like *pocha-pocha* 'chubby', *puyo-puyo* 'fat', and *puku-puku* 'puffing up' (Iida and Funakura, 2024). This suggests that the method used to disambiguate sharp and round pseudowords and images may pick up on relationships between semantic concepts and word forms—being heavily entangled with the choice of ground-truth adjectives—rather than capturing true sensory mappings in languages. Moreover, the vectors used to assign a geometric or phonetic score to a pseudoword or image must be sufficiently dissimilar, for which Iida and Funakura (2024) reported that this was not the case. The contradictory findings between Japanese and English VLMs highlight the need for a different approach that aligns with human experiments on cross-modal associations.

A key ingredient of human bouba-kiki experiments is that tests are centred around specifically designed pairs of visual images that minimally differ, but with one more rounded and one more jagged version, as shown in Figure 1. For example, Maurer et al. (2006a) presented a pair of jagged/round images along with a pair of bouba/kiki-like words and participants were asked to pair them up in the most fitting way. Other studies employed more stringent tests, as in Ćwiek et al. (2022), where only the images were presented side by side, and participants had to choose the best-fitting image after listening to only one of the spoken words at a time. Finally, Nielsen and Rendall (2013) presented single images and asked participants to generate novel pseudowords to match the shapes. In all of these cases, humans exhibit a strong bouba-kiki effect. Inspired by these studies, Verhoef et al. (2024) used images from existing human experiments and explored four prominent VLMs on carefully designed image-to-word matching tasks. They directly used model probabilities generated by VLMs to match specific pseudowords with images as a measure of preference and found limited evidence for the bouba-kiki effect. Two out of four models (CLIP and GPT-4o) exhibit moderate alignment with human-like associations, but only in some of the tests they conducted, and not consistently. The study concludes that cross-modal associations in VLMs are highly dependent on factors such as model architecture, training data, and the specific test used.

Others have also investigated the bouba-kiki effect and other cross-modal associations, such as a relation between perceived size and vowels (Loakman et al., 2024) and understanding shitsukan terms (a Japanese concept that captures the sensory essence of an object) (Shiono et al., 2025). However, none of these consistently find a resemblance between the human and model associations. Tseng et al. (2024) used the embedding method from Alper and Averbuch-Elor to test sensitivity to sound-symbolic associations in audio-visual models and report that these models capture sound-meaning connections akin to human language processing. Yet, this method again seems to rely on the relationships between semantic concepts and word forms, as was mentioned before. Given the conflicting evidence in this domain, we revisit the bouba-kiki effect through a thorough investigation using two versions of CLIP and a wide variety of prompts. This approach introduces a novel method for testing it using visual interpretability methods and directly compares the results with similar findings from human studies.

# 3 Methodology

Cross-modal associations are assessed by prompting two different versions of CLIP (Radford et al., 2021), specifically, a ResNet-50 and a ViT version. We deem this as reasonable since state-of-the-art VLMs such as Molmo (Deitke et al., 2024), LLaVA (Liu et al., 2023), BLIP2 (Li et al., 2023), and InternVL (Chen et al., 2024) commonly use pre-trained frozen vision models and train lightweight alignment layers to align visual features with existing linguistic embeddings present in (large) language models (Liu et al., 2024). The frozen ViT version of CLIP is particularly often the basis for many current state-of-the-art VLMs (BLIP2, Molmo, InternVL, LLaVa, i.a.). If CLIP, as a backbone, does not consistently exhibit human-like cross-modal associations, it is difficult to imagine how additional alignment layers can capture human-like representations without extensive fine-tuning. Moreover, CLIP enables us to extend existing approaches with an interpretability-based methodology that cannot be applied to proprietary models.[2]

**Linguistic inputs**  Models are probed through combining images and labels, the latter of which are pseudowords originating from four different sources. First, two sets of 'original' labels are used, which have been traditionally used the most in human studies (*bouba-kiki* and *maluma-takete*), allowing for explicit comparison of human results with those of VLMs. Second, we borrow English adjectives from Alper and Averbuch-Elor (2023) that are 10 synonyms of 'sharp' and 'curved'. These serve as a baseline informing us whether the models can, in principle, make correct cross-modal associations. This differs from Alper and Averbuch-Elor (2023), who use adjectives to create a vector to calculate a geometric score. Third, two-syllable labels were constructed following Nielsen and Rendall (2013), using previously established cross-modal patterns in English. Sonorant consonants (M, N, L) and rounded vowels (OO, OH, AH) tend to match curved shapes, while plosive consonants (T, K, P) and non-rounded vowels (EE, AY, UH) align with jagged shapes. Combining these yields 36 syllables (e.g., loo, nah, kee, puh), categorised into four types: sonorant-rounded (S-R), plosive-rounded (P-R), sonorant-non-rounded (S-NR), and plosive-non-rounded (P-NR). These were paired into two-syllable pseudowords, loosely replicating the task humans did in Nielsen and Rendall (2013). For analysis, we focus on 'pure' pseudowords that are either fully round-associated (S-R-S-R) or fully sharp-associated (P-NR-P-NR), yielding 162 labels. Fourth, VLMs are tested on pseudowords generated using the method from Alper and Averbuch-Elor (2023). Each pseudoword follows a three-syllable structure, where consonants (P, T, K, S, H, X, B, D, G, M, N, L) are combined with vowels (E, I, O, U, A).[3] Pseudowords repeat the first syllable at the end (e.g., 'kitaki', 'bodubo'). Only 'pure' items—composed entirely of syllables from a single class—are used, excluding mixed forms like 'kiduki'.

The linguistic inputs are comprised of a label of interest—which should induce certain preferences—and a prompt ('The label for this image is <label>') such that embedding the label happens at the sentence level and is closer to the models' natural objective. Importantly, for each image, we *only* differ the pseudowords in question, so variation in probability must be a result of the pseudoword. Provided that the pseudowords are generated anew, it is unlikely that the models encountered them during training. Ten different prompts are used (see Appendix A) to ensure that the results are robust and not an artefact of peculiarities in the prompts. These prompts are sourced from Verhoef et al. (2024), Alper and Averbuch-Elor (2023), or are newly created such that half of the labels appear as a noun, and the other half as an adjective.

**Visual inputs**  The images fed to the models are either curved or jagged. Some of these images are sourced directly from previous works involving human participants (Köhler, 1929, 1947; Maurer et al., 2006a; Westbury, 2005b), while others are specifically generated to test cross-modal associations in VLMs. These were inspired by the method described in Nielsen and Rendall (2013) and already used by Verhoef et al. (2024). The generated images were created by randomly distributing points within a circle and then connecting them sequentially using curved lines for curved images and straight line segments for jagged images. Hence, this method generated image pairs that subtly differ *only* in features that are seemingly important for cross-modal effects. This differs from Alper and Averbuch-Elor (2023), who generated images using Stable Diffusion, leaving less experimental

---

[2]The source data and code are available at `https://osf.io/gqsv6/`

[3]The letter A is included per Alper and Averbuch-Elor (2023) despite that it is typically not regarded to evoke cross-modal associations in humans.

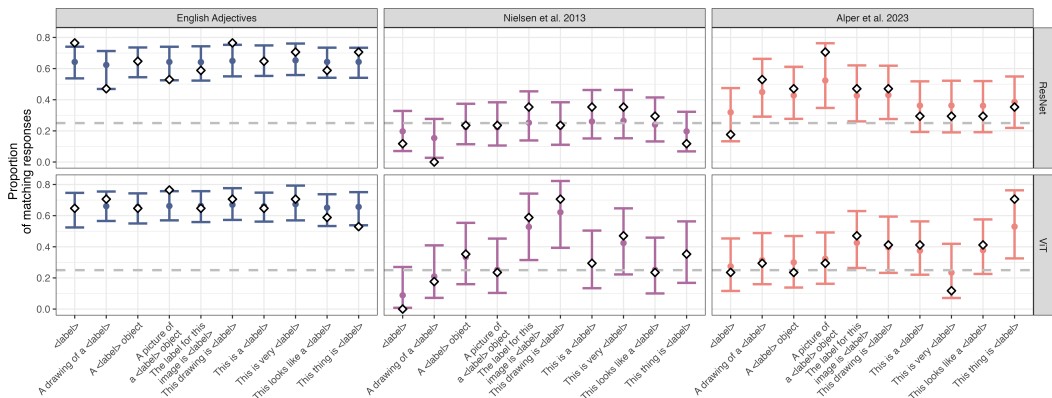

Figure 2: The proportion of congruent responses for matching both images of an image pair correctly ($Match = 1$). A result in which models consistently match images above chance (the grey dashed line) across prompts would suggest the presence of cross-modal associations. This is only the case for the English adjectives, which function as a baseline. Model: $Match \sim 1 + Word\_type + (1 + Word\_type | Prompt)$. Diamonds are descriptive means, and the dots are posterior means.

control and potentially distorting the understanding of which features cause the observed effects. An overview of all image pairs is displayed in Appendix B.

**Analyses**   All our analyses use Bayesian Regression Models as implemented in the *brms* package (Bürkner, 2021) in R (R Core Team, 2024). We fit models (using 4 chains of 4000 iterations and a warm-up of 2000) to predict the proportion of correct guesses given a *Word_type* with fixed effects for *model*, *prompt*, *image pair*, or *label pair*. The exact model formulas are displayed under each figure. The interpretation of our visualisation is straightforward: an effect is significant if the posterior means *and* their credible intervals are above the chance level.

## 4   Probing through probabilities

To assess the preferences of the ViT and ResNet versions, we extract probabilities for all possible labels (i.e., syllables and pseudowords) conditioned on an image. For each image, we consider the label with the highest probability to be the model's preference (shown on the left in Figure 1). This is comparable to how Nielsen and Rendall (2013) tested for human preferences. While using the probability data for all labels would have been possible, previous work demonstrated that using only the best-matching label yielded a small bouba-kiki effect, whereas using all probabilities did not (Verhoef et al., 2024); as such, we use the most promising method. Importantly, we follow Ćwiek et al. (2022), who rightfully treated human responses as bouba/kiki-congruent when participants matched bouba to a round shape, and crucially, also matched kiki to a spiky shape. In our work, this means that an image pair of a sharp *and* round shape must be matched with a congruent label.

**Results**   We begin by considering the English adjectives, which serve as a baseline and do not inform us about the bouba-kiki effect. Figure 2 reveals that these are successfully matched to images with round or sharp features. This result is consistent across prompts and is also somewhat expected. Yet, a performance of roughly 80% across prompts and models could also be considered low given the clear-cut distinctions between images and adjectives. The primary labels of interest (i.e., the pseudowords) can not be robustly and correctly matched to images with corresponding features by either model version. Despite some variability across prompts and models, with some combinations leading to above-chance performance (e.g., ViT, Nielsen and Rendall's pseudowords, and *This drawing is <label>* or ResNet, Alper and Averbuch-Elor's pseudowords, and *A picture of a <label> object*), the descriptive means are mostly at chance level. Hence, we conclude that neither model displays clear cross-modal preferences. Interestingly, the individual modalities are, in principle, separable by both models (A.1 and B.1), suggesting that the difficulty lies in combining information from multiple modalities. This is also visible when looking into which labels are most commonly chosen, revealing that model predictions do not differ much even though they are presented

with different images (Appendix C). For the experimental pseudowords, models seem to rely on a few frequently selected labels (the percentage of unique labels for ResNet and ViT is $\approx 35\%$). This strengthens our observation that no robust syllable-level cross-modal associations influence predictions.

## 5   Beyond behavioural observations

In addition to using probability data, we are interested in knowing whether the predictions are wrong for the right reasons, i.e., do they fail to attend to curved or jagged regions when presented with our images? As such, we further analyse the behaviour and preferences of models when associating an image with a label, using Grad-CAM, a technique from the model interpretability literature (Selvaraju et al., 2020). Grad-CAM offers a visual explanation of a model's prediction by calculating the gradients of the target class score. This involves using the cosine similarity between the label and image embeddings in relation to the feature maps from the final convolutional layer or the last attention block. In the case of ViT, we applied Grad-CAM to the last attention layer, which retains spatial information via its attention heads. We specifically focused on the attention from the [CLS] token to the image patches, similar to the approach in Caron et al. (2021) and following public implementations of Grad-CAM (Zakka, 2021; Mamooler, 2021; Chefer et al., 2021). Specifically, to identify which image regions contributed most to the decision, we computed the gradients of the target score with respect to the attention weights. These are averaged across heads, and we then extract the attention weights from the [CLS] token to the image patches by removing the [CLS] column. This is reshaped into a 2D spatial grid that acts as a feature map. The values in these feature maps (typically displayed as a heatmap) can be interpreted as the importance or contribution of a specific region in the image to the model's prediction (see Appendix D for some visual examples).

This technique enables us to very closely mimic the cross-cultural study conducted by Ćwiek et al. (2022) in which human participants were shown the two classical bouba-kiki shapes and listened to the spoken words bouba or kiki. Hereafter, they selected which of the two shapes they thought corresponded to the word. To simulate this experiment with VLMs, we concatenate all image pairs into a single image containing a curved and jagged shape (n=17). We then identify, for each label—which defines the expected target—which image region (left or right) receives the most attention from a model using Grad-CAM (see Figure 1). While the generated heatmaps visually indicate attention patterns, we quantify attention by summing the attention allocated to each curved and jagged part of the image. Taking the total attention allocated to each part, rather than focusing on very specific regions, aligns with how humans perceive faces, objects, and words holistically (Taubert et al., 2011; Zhao et al., 2016; Wong et al., 2011). Nevertheless, we also experimented with entropy and centroid-of-attention as quantifying measures, but none of these changed the results meaningfully (Appendix D. We compare the sum of importance values in the expected region for a given pseudoword within a particular category with the sum of importance values in the non-expected region for each image and text label. Based on this comparison, we compute the percentage of trials in which the model consistently focuses more on the expected region than on the non-expected region (as shown on the right in Figure 1). The way images are concatenated is balanced such that the target appears eight or nine times on the left or right. Doing this eliminates a potential model bias toward objects on the left or right. Initial analyses across models, prompts, labels, and images revealed that both models are relatively consistent in their predictions, regardless of the target location, with ResNet being consistent for $77.0\%$ of the predictions and the ViT variant $73.5\%$ (Appendix D).

**Comparing VLMs with humans**   Using Grad-CAM, we compare the performance of both models to that of English-speaking participants as reported by Ćwiek et al. (2022). While they only investigated the well-known label pair bouba-kiki, another pair, maluma-takete, combined with the two images shown in Figure 1, was first described by Wolfgang Köhler (Köhler, 1929, 1947) and sparked wider interest in cross-modal associations. We use only these two label pairs here, since the larger pseudoword set does not contain paired labels. However, our image generation process, which only varies how points are connected, allows evaluating across all image pairs, beyond just the original images. Specifically, we evaluate whether models attend more to the expected region (e.g., the sharp shape for 'kiki' and 'takete' or the round shape for 'bouba' and 'maluma'). A prediction is considered congruent only if both labels are correctly matched to their corresponding shapes. This setup closely mirrors human experiments and, for the bouba-kiki label pair, enables direct comparisons.

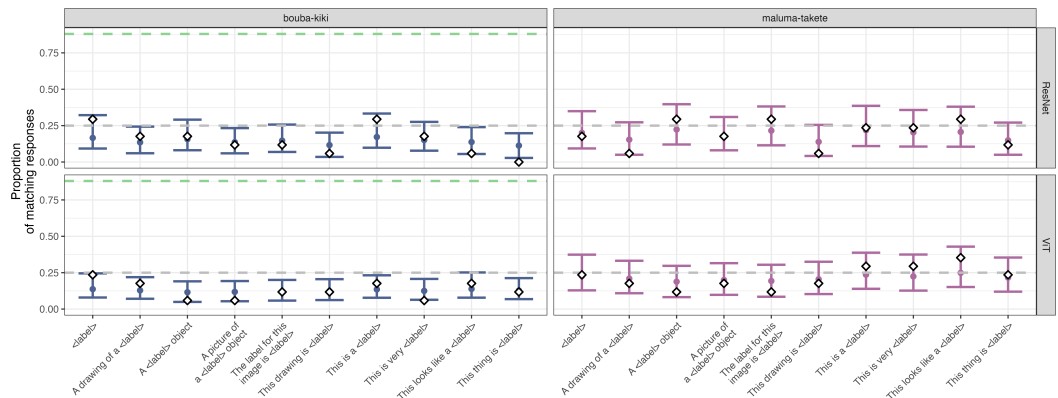

Figure 3: The proportion of correctly matched responses for both labels of a label pair ($Correct = 1$) given an image pair using Grad-CAM. The green line indicates human 'performance' reported in (Ćwiek et al., 2022). The grey line shows the chance level (25%) as the model must map 'bouba' *and* 'kiki' correctly. Model: $Correct \sim 1 + LabelPair + (1 + LabelPair|Prompt)$. Diamonds are descriptive means, and the dots are posterior means.

Strikingly, neither CLIP model consistently maps both labels to their intended shapes at a human-like level (Figure 3); in fact, performance does not reliably exceed chance. These results contradict previous claims that vision–language models exhibit human-like cross-modal associations (Alper and Averbuch-Elor, 2023), and instead reinforce earlier findings showing no such effect. If there is any situation in which an effect would be expected, it would be here since these classic word pairs are most dominantly present in the data. Yet, these results show that merely learning *about* an existing cross-modal effect from data distributions is different from having a mechanistic preference for cross-modal associations, which should not come as a surprise.

**Analysing pseudowords** Extending the analyses beyond the original word pairs, we now assess all pseudowords to examine whether CLIP displays a fundamental association between syllables and shapes. The proportion of pseudoword-label responses displayed in Figure 4 reveals that there is again some variability across prompts, but model preferences are relatively consistent. Using this method, the ResNet variant displays a general preference to attend to round shapes across all word types (note that not attending to a jagged shape if the label is of the jagged category means the model attends

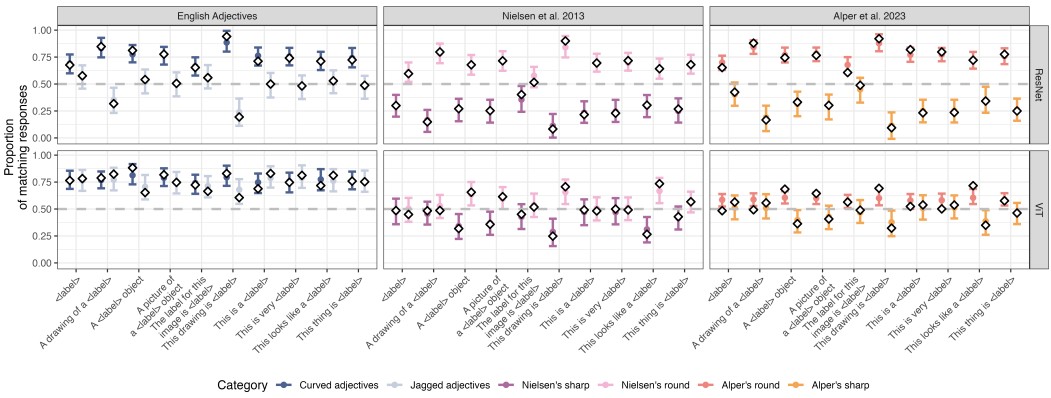

Figure 4: The proportion of correct matches (*CorrectProportion*) given a word type and category (i.e., curved or sharp). A cross-modal association is indicated when a model consistently matches images above chance (grey line) for both categories—observed only with ViT and the English adjectives used for comparison. Model: $CorrectProportion \sim 1 + Word\_type + Category + (1 + Word\_type + Category|Prompt)$. The diamonds are descriptive averages, and the dots are posterior means.

to the round shape). Provided that this is also the case for the English adjectives, it is unsurprising that the pseudowords do not display a bouba-kiki effect. The ViT variant is relatively consistent and can reliably distinguish shapes for English adjectives. Yet, using experimental pseudowords collapses performance to chance levels, indicating the absence of human-like associations between shape features and syllables or characters. A qualitative inspection of several attention patterns provides additional evidence against the presence of cross-modal associations in CLIP. This reveals that attention patterns primarily do not focus on sharp edges or round attributes of images, but instead mainly focus on the centres of shapes or background areas (Appendix D). The latter is similar to earlier findings by Darcet et al. (2024) who found that ViT networks create artefacts at low-informative background areas for computational purposes rather than describing visual information. Both observations are in contrast with what, at its core, is required for a bouba-kiki-like effect.

# 6 Discussion and conclusion

This work investigated whether vision-and-language models, specifically two versions of CLIP (ResNet and ViT), exhibit human-like patterns of cross-modal association, as reflected in the bouba-kiki effect. We aimed to rigorously evaluate the presence or absence of this phenomenon in model behaviour, using two methodological approaches closely aligned with human experiments and introducing the use of interpretability tools in this domain to probe internal representations. Crucially, we posit that we can only speak of a consistent model preference when results across both methodologies are congruent with human-like preferences. Our first experiment, which used image-label probabilities to gauge model preference, showed no clear cross-modal preference for either model. The second experiment utilised Grad-CAM to simulate experimental conditions in human studies. Here, the ResNet variant, which showed an overall preference for round shapes, did not demonstrate behaviour consistent enough to qualify as a bouba-kiki effect by human standards. The ViT-based version of CLIP, widely used as a foundational component in many current multimodal systems, proved capable of matching shapes and English adjectives, but failed to do so for pseudowords, which have even weaker alignment than ResNet. Contrary to prior work suggesting VLMs may encode bouba-kiki-like associations (Alper and Averbuch-Elor, 2023), our findings reveal little to no evidence that CLIP models, under these experimental conditions, exhibit consistent human-like mappings between pseudowords and visual shape features. This lack of alignment suggests that, despite their impressive performance across many downstream vision-language tasks, models like CLIP do not appear to represent cross-modal associations in a cognitively grounded manner internally. Together, this raises questions about how cross-modal grounding is encoded or inherited in many larger state-of-the-art architectures.

Our findings complicate earlier claims about the presence of sound-symbolic associations in model embeddings. While Alper and Averbuch-Elor (2023) found strong evidence for a bouba-kiki effect using embedding similarity scores in CLIP and Stable Diffusion, Iida and Funakura (2024) used the same method with Japanese VLMs, but failed to pick up robust bouba-kiki-like mappings, even though Japanese is rich in sound-symbolism (Dingemanse, 2012). Instead, the embedding similarity scores seem to reflect language-specific regularities rather than general cross-modal preferences. Our approach deliberately avoided this by mirroring psycholinguistic testing paradigms with carefully controlled pairs of novel images. Corroborating other recent studies (Verhoef et al., 2024; Loakman et al., 2024), this empirically grounded probing method yielded results opposite to those of other studies, showing no evidence of a bouba-kiki effect. Whereas Verhoef et al. (2024) report limited, though not consistent, evidence for a bouba-kiki effect, our work differs since it uses a more comprehensive set of pseudowords, amending their analyses and suggesting that there is no general effect. We also ran the same tests using English adjectives, and in contrast to pseudowords, the models are able to make the expected mappings in that case. Furthermore, by employing Grad-CAM in the domain of cross-modal associations, we examined whether models visually attend to shape features that correlate with pseudoword form profiles. This enabled us not only to assess whether the models exhibit human-like cross-modal associations, but also to determine whether they do so for the right reasons by probing the decision-making pathways within the models. They largely did not attend to the expected shape features, which presents further evidence for the claim that VLMs may lack human-like cross-modal representations.

What causes this misalignment between humans and VLMs? The broader implication is that current VLMs, even those trained on massive paired datasets, lack a key component of human-like multisen-

sory understanding: grounded, flexible, and intuitive mappings between sensory modalities. This is perhaps not surprising given the lack of embodied interaction in their training and reliance on statistical co-occurrence rather than perceptual salience. Still, it can be argued that the co-occurrences in the training data are full of human-like preferences, including cross-modal associations. So while we may expect VLMs to pick up on these associations, their internal mechanisms inherently differ from those of humans and do not necessarily learn the same preferences from aligned image-text pairs. Previous work on visual grounding (Jones et al., 2024), and shortcut learning, i.e., solving a test using unexpected non-human-like shortcuts (Schwartz and Stanovsky, 2022; Mitchell and Krakauer, 2023) argued similarly. However, interestingly, we observed that the individual modalities are, in principle, separable. This suggests that CLIP's training objective (i.e., a sentence-level contrastive loss) and its method for aligning text and image modalities (i.e., cosine similarity) are the culprits. Neither seems to specifically promote learning relations between visual features and phonetic elements in language.

Another straightforward difference between humans and VLMs is tokenisation. Unlike humans, who perceive words holistically (Wong et al., 2011), tokenisation can distort word representations, reducing the potential for human-like associations with visual shapes. Inspection of the tokenised pseudowords, however, reveals that this does not occur frequently ('Bouba' becomes 'bou' and 'ba', 'kiki' is one token, 'sepise' becomes 'sep', 'ise', and 'kaykuh' becomes 'kay', 'ku', 'h'. See section A.2 for more examples.) Most phonological structures are preserved that could match shape features. However, tokenisation may still split some pseudowords in ways that wouldn't evoke the expected cross-modal associations in humans (e.g., 'H' in 'OH' might elicit jagged rather than curved associations). In the cases where tokenisation breaks pseudowords into syllables that also show no bouba-kiki effects in humans, the set of alternative pseudowords is large enough to contain options that could invoke these associations. If models had a genuine preference in the direction of a bouba-kiki effect, they would prefer those pseudowords that are tokenised as complete syllables, and retain the potential cross-modal association of interest, over those split into non-syllables in which the association may be corrupted. Our results show this is not the case. As such, the potential value of developing models with shared preferences between humans and machines remains significant. Establishing alignment in human and machine understanding of (visual-auditory) form-meaning mappings and mutual understanding can enhance their interactions (Kouwenhoven et al., 2022), helping in creating AI systems that resemble how humans process and communicate meaning. A potentially rich line of future work includes systematically exploring how design choices in VLMs affect the emergence of cognitively meaningful representations and consistently incorporating cross-linguistic studies to account for cultural variation (as demonstrated by Iida and Funakura (2024)).

Our work has some notable limitations. First, we only test on CLIP versions. We deliberately chose to focus on the CLIP model instead of more capable proprietary models (GPT-4o and Gemini2-Flash, i.a.), since they are not sufficiently transparent and thus do not help advance our understanding of their internal mechanisms. Nevertheless, it remains interesting to further unravel cross-modal associations in more contemporary models. Although they often use frozen ViT variants of CLIP, the alignment layer between CLIP's embedding and the language model's embedding may learn preferences that resemble those of humans. While the original bouba-kiki effect is rooted in sound symbolism, our work—and that of others (e.g., Alper and Averbuch-Elor, 2023; Loakman et al., 2024; Iida and Funakura, 2024)—relies on text, which may influence the outcomes. Yet, prior human experiments present pseudowords both acoustically and in written form (e.g., Nielsen and Rendall, 2013), making it challenging to disentangle orthographic from auditory contributions. Moreover, Cuskley et al. (2017) showed that auditory presentation cannot eliminate orthographic effects in literate participants, as characters are strongly associated with particular sounds. These associations are non-arbitrary: writing systems often employ iconic strategies, in which characters representing rounded sounds (rounded vowels, sonorants) tend to have curved shapes (Koriat and Levy, 1977; Turoman and Styles, 2017). Given this tight coupling between sound and orthography, correspondences between vision and spoken words should be highly correlated with vision-text correspondences. This makes it unlikely that audio-visual models would show more substantial bouba-kiki effects than text-based models. Nonetheless, exploring audio-visual models (Tseng et al., 2024) remains a valuable direction for understanding how different modalities contribute to cross-modal associations.

Overall, our results reinforce the importance of interpretability and cognitively inspired evaluation when assessing model performance in cross-modal reasoning. If VLMs are to serve as truly intuitive agents in real-world human-machine interactions, they must not only succeed on benchmark datasets but also exhibit a more human-like understanding of abstract, grounded concepts.

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

# A  Linguistic inputs

The experimental inputs are tested across ten different prompts to assess robustness. They originate from earlier investigations (Alper and Averbuch-Elor, 2023; Verhoef et al., 2024) and are extended with additional prompts such that the *label* occurs equally frequently as a noun and adjective (Table 1).

| Prompt | Word type | Origin |
|---|---|---|
| *The label for this image is <label>* | Noun | Verhoef et al. (2024) |
| *This is a <label>* | Noun | new |
| *A drawing of a <label>* | Noun | new |
| *This drawing is <label>* | Adjective | new |
| *This thing is <label>* | Adjective | Alper and Averbuch-Elor (2023) |
| *A <label> object* | Adjective | Alper and Averbuch-Elor (2023) |
| *A picture of a <label> object* | Adjective | Alper and Averbuch-Elor (2023) |
| *<label>* | Noun | Alper and Averbuch-Elor (2023) |
| *This looks like a <label>* | Noun | new |
| *This is very <label>* | Adjective | new |

Table 1: Different linguistic inputs are used to test our models for cross-modal associations.

## A.1  Textual embeddings

To make correct associations, the models must, at a minimum, be able to disambiguate labels (real and non-words) from each other. If they fail to do so, it is presumably also impossible to link certain words and their linguistic features to shape-specific features. Figure 5 displays a t-SNE visualisation of the models' embeddings and reveals that they, in principle, should be able to disambiguate labels from different categories within a word type.

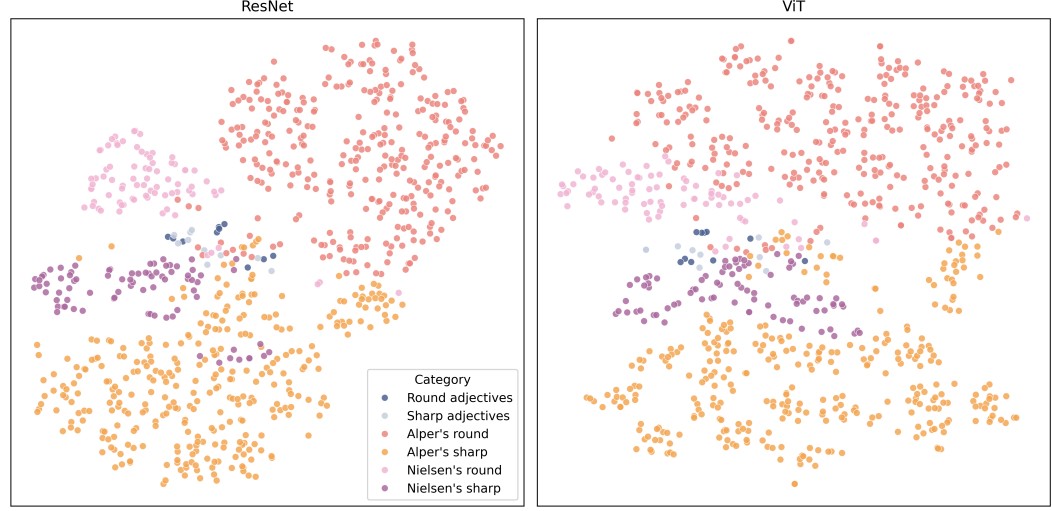

Figure 5: t-SNE plot showing how the language models of different CLIP variants interpret labels from different categories. The colour shades indicate which word type a label in a category belongs to. In order to correctly match labels to images with shape-specific features, a model must be able to discriminate word types between labels of the same category. This is clearly possible. This plot shows the embeddings for the prompt: *The label for this image is <label>*. Different plots result in similar distributions.

## A.2  Tokenisation

To determine whether byte-pair encodings or our pseudowords break phonological structures, we qualitatively assess a set of tokenised examples. These pseudowords are parsed as elements of the

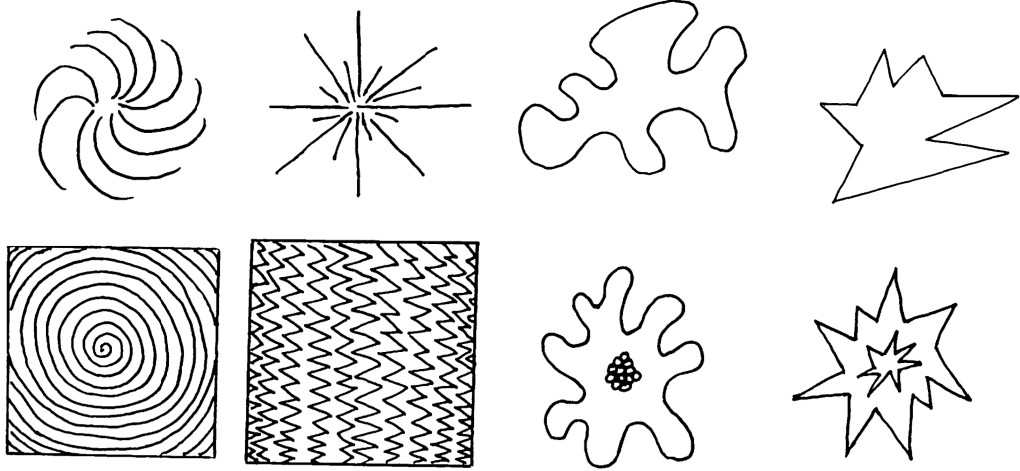

Figure 6: Image pairs from (Maurer et al., 2006b)

sentence prompts, but we only focus on the pseudowords themselves. The list below reveals that most of the target words are tokenised in a way that preserves at least some phonological structures that could be matched to shape features.

- Bouba → 'bou' and 'ba'
- Kiki → 'kiki'
- Takete → 'take', 'te'
- Maluma → 'mal', 'uma'
- Xehaxe → 'xe', 'ha', 'xe'
- Lohlah → 'loh', 'lah'
- Sepise → 'sep', 'ise'
- Kaykuh → 'kay', 'ku', 'h'
- Loomoh → 'loom', 'oh'
- Mohmah → 'moh', 'mah'

## B   Visual inputs

The full set of images with visual shapes that were used in the experiments is shown here. Besides the original image pair from Köhler (1929, 1947), we used four image pairs from Maurer et al. (2006b), displayed in Figure 6, four from (Westbury, 2005a, ; Figure 7), and eight generated pairs following the method described by Nielsen and Rendall (2013) and used in (Verhoef et al., 2024, ; Figure 8;). For each image pair, the Curved version is displayed on the left and the Jagged version on the right.

### B.1   Visual embeddings

Similar to the textual embeddings presented before, Figure 9 displays a t-SNE visualisation of the models' visual embeddings. It is visible that the models should, in principle, be able to disambiguate our target images based on their condition, even when they only differ in how the randomly distributed points are connected.

## C   Unique labels

To test whether the models in the first experiment (section 4) change their predictions according to a set of images, we report the ratio of unique labels in Table 2. In this case, we are not concerned with

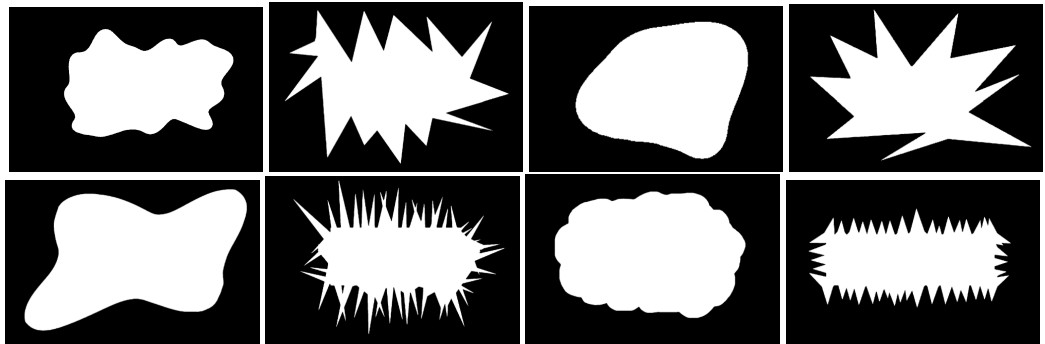

Figure 7: Image pairs from (Westbury, 2005a)

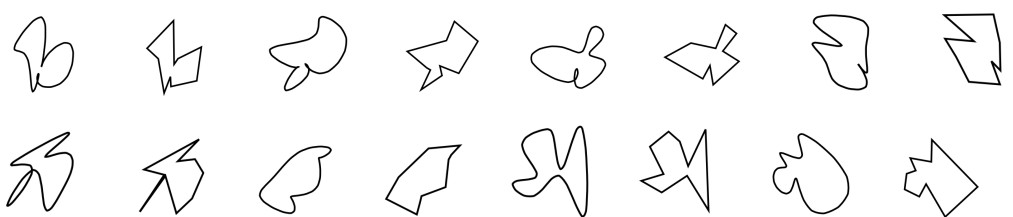

Figure 8: Newly generated image pairs

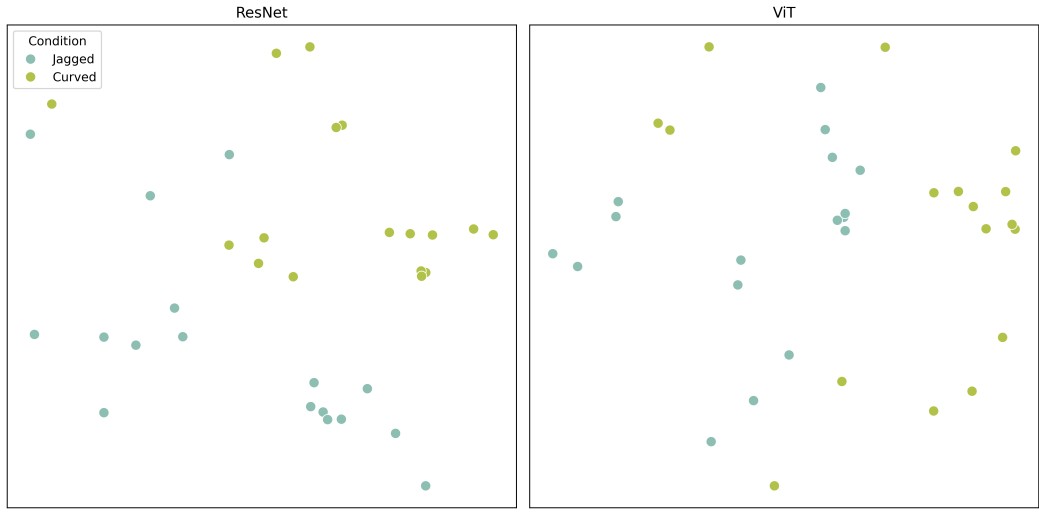

Figure 9: t-SNE plot showing how the vision models of different CLIP variants interpret images from conditions. The colour shades indicate which image condition an image belongs to. In order to correctly match labels to an image, a model must be able to discriminate the images based on their shape. This is clearly possible.

| Model | Word type | Ratio |
|-------|-----------|-------|
| ResNet | Initial words | .675 |
| ResNet | English adj. | .275 |
| ResNet | Nielsen et al. | .329 |
| ResNet | Alper et al. | .444 |
| ViT | Initial words | .800 |
| ViT | English adj. | .320 |
| ViT | Nielsen et al. | .331 |
| ViT | Alper et al. | .450 |

Table 2: The average ratio of unique labels chosen for each image (n=34) across different prompts (n=10) in different label sets. The latter means that we divide the unique labels by the length of the set of possible labels to gauge diversity. A high ratio indicates that the corresponding model assigned different labels to different images.

whether the models make predictions that align with the bouba-kiki effect, but rather with the diversity of their predictions. We calculate the ratio of unique labels for each word type across all images and the ten prompts. These ratios should, in a scenario where cross-modal associations are present, be high, and the number of correct trials, as seen in Figure 2, would be above chance. However, it is clear that the models somewhat change their predictions (mainly for the initial pseudowords) when they are conditioned on different images, but mostly *collapse* onto the same labels for different images. Table 2 confirms this by showing that there is only slight variation among picked labels for each prompt. Given that responses are only counted as congruent when both predictions (i.e. the prediction for a jagged and curved shape) are correct, this explains why we find that both models perform at chance levels. A qualitative example is provided in Table 3, which shows the unique labels and their ratios for a single prompt. The model and word type that are most affected by different images (i.e. have a high ratio of unique words) also exhibit the strongest bouba-kiki like effect Figure 2. This appears to happen even though, in the case of both ViT and ResNet, and Alper and Averbuch-Elor labels, the majority of the labels would be associated with sharp images by humans.

| Model | Word type | Ratio | Uniquely chosen labels |
|-------|-----------|-------|------------------------|
| ResNet | Initial words | .250 | bouba |
| ResNet | English adj. | .300 | angular, circular, curved, prickly, rotund, spiky |
| ResNet | Nielsen et al. | .412 | kaykee, kuhpay, kuhpee, kuhpuh, lahmoo, lohloh, lohmah, lohmoo, mohmah, nahmoo, nohmoo, paykuh, peepay, teepee |
| ResNet | Alper et al. | .471 | kehake, kehike, ladula, lunulu, malama, nulunu, paxapa, pihapi, pikepi, pikipi, pixapi, sepise, tatata, tekete, xaxixa, xehexe |
| ViT | Initial words | .500 | bouba, takete |
| ViT | English adj. | .250 | angular, circular, curved, rotund, spiky |
| ViT | Nielsen et al. | .412 | keekay, kuhtay, lahmoo, lohlah, loomoh, mahloh, mahnoh, mohloo, mohnoh, nohloh, noonoo, taypay, taypuh, teepee |
| ViT | Alper et al. | .676 | dododo, hexehe, kixiki, lamola, lubalu, lunulu, mamuma, monomo, mubomu, patipa, pisipi, pixapi, sahisa, satasa, tehite, texate, xahexa, xakixa, xasixa, xehexe, xikexi, xipaxi, xisixi |

Table 3: The set of unique labels chosen across all images (n=34) and its ratio for an example prompt ('The label for this image is <label>'). The latter means that we divide the unique labels by the length of the set of possible labels to gauge diversity. A high ratio indicates that the corresponding model assigned different labels to different images.

# D  Grad-Cam visualisations, consistency, and additional results

Figure 10 provides example visualisations of the attention patterns for different randomly selected image pairs and randomly selected labels across both models. Here, we sampled different image pairs for the models as to give a complete view of the images used. The pseudowords are consistent for the columns. The images reveal that models mostly attend to the centres of shapes and/or focus on non-informative background areas. The latter is also described by Darcet et al. (2024). Although this is a small sample, it is clear that the target labels do not steer models towards shape-specific features.

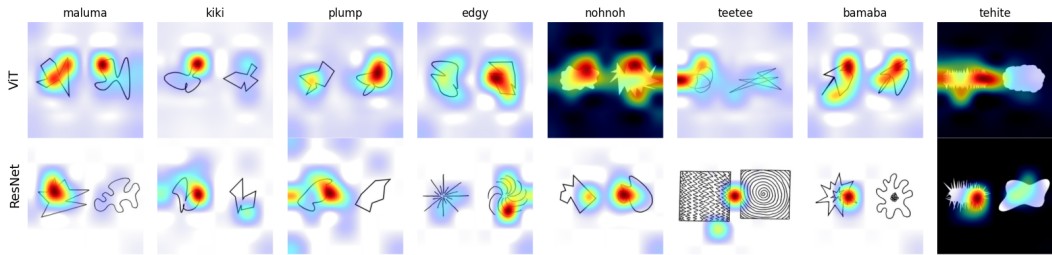

Figure 10: Visualising the attention pattern for the text prompt: 'The label for this image is <label>' in both models.

## D.1  Prediction consistency

To test whether the models in the second experiment (section 5) change their predictions resulting from different shape positions, we report the ratio of consistency in Table 4. In this case, we are not interested in whether the models make predictions that align with the bouba-kiki effect, but rather in the consistency of their predictions. Both models are rather consistent in the mappings they make between labels and shapes.

## D.2  Quantifying model preference

The results presented in section 5 utilise the sum of attention values to quantify the models' preferences, as this aligns with humans' holistic perception (Taubert et al., 2011; Zhao et al., 2016; Wong et al., 2011). Though artificial models may show their preference differently. To this end, we additionally experimented with the entropy and centroid-of-attention as quantifiers of model preference.

Comparing the predictions (averaged over all images, labels, and prompts) quantified by the centroid of attention with the sum of attention, we find that the predictions using the centroid of attention strongly overlap with those resulting from the sum of attention (ViT: 85.6% and ResNet: 88.0%). Given this considerable overlap, it is not surprising that the results remain highly similar when using

| Model | Word type | Ratio |
|---|---|---|
| ResNet | Initial words | .799 |
| ResNet | English adjectives | .761 |
| ResNet | Nielsen et al. | .758 |
| ResNet | Alper et al. | .765 |
| ViT | Initial words | .723 |
| ViT | English adjectives | .749 |
| ViT | Nielsen et al. | .722 |
| ViT | Alper et al. | .739 |

Table 4: The ratio of consistently attending to the same shape in a different position when presented with the same label. Values are averages across image pairs, prompts, and labels. A high ratio indicates that the corresponding model consistently focuses on the same shape, even when the target location is different.

| Model | Sum | Entropy | Centroid |
|-------|-----|---------|----------|
| ResNet | .519 | .488 | .515 |
| ViT | .522 | .514 | .515 |

Table 5: The correctness of all model prediction types using different quantifications averaged across all images, labels, and prompts.

this alternative method (Table 5. Yet, for the entropy of attention, the predictions (where the image half with the lower entropy acts as the model's preference) for ViT overlap strongly (80.2%) but not for ResNet (21.1%). This entropy method, despite yielding different predictions, still results in a slightly lower number of correctly predicted images, with only $48.8\%$ of the shapes correctly identified, compared with $51.9\%$ for the quantification method that uses the sum of attention. So if we were to use entropy-based predictions, the performance would be even worse than predictions based on the sum of attention.

