# OpenReview forum: "Cross-modal Associations in Vision and Language Models: Revisiting the Bouba-Kiki Effect"
_NeurIPS.cc/2025/Conference — NeurIPS 2025 poster_

### Official Review · Reviewer_74nD · 2025-06-08

**Clarity:** 3
**Significance:** 2
**Originality:** 4
**Rating:** 4
**Confidence:** 4

**Summary:**

This paper addresses previous inconsistencies in evaluating whether VLMs can capture cross-modal associations. It presents a comparative analysis of two CLIP variants across two experiments, using a more comprehensive and diverse set of pseudowords. To probe the models’ behavior, the authors apply a gradient-based explainability method to assess the influence of visual features. Their findings reveal that the cross-modal associations learned by VLMs diverge significantly from those observed in humans.

**Questions:**

**Reflection on the reasons behind misalignment**: the results showing that English adjectives lead to consistent cross-modal associations are compelling, especially since they suggest that VLMs can connect semantic meanings to visual features. However, this pattern does not extend to sound-symbolic associations like bouba-kiki. Do the authors attribute this discrepancy to the nature of CLIP's training data? Since image captions rarely reflect symbolic or phonetic associations, it seems plausible that such mappings are simply not represented during pretraining. Could the authors elaborate on this potential limitation and whether alternative datasets or finetuning could help reveal such associations?

**Clarification on tokenization effects**. The claim that tokenization hinders the analysis of sound-symbolic effects deserves further clarification. While tokenization may segment words, syllabic structures are often preserved, especially in pseudowords designed to evoke phonetic patterns. Therefore, it is not immediately clear how tokenization alone would disrupt bouba-kiki-like associations. Could the authors specify more precisely how tokenization interferes with these effects?

**Ethical Concerns:**

["NO or VERY MINOR ethics concerns only"]

**Final Justification:**

We believe that this score is adapted to the quality of the work, and our concerns were addressed. I would keep my current ratings.

**Limitations:**

yes

**Paper Formatting Concerns:**

No concern

**Quality:**

3

**Strengths And Weaknesses:**

Strengths:

- This work offers a novel and systematic investigation into cross-modal associations between language and vision. It clearly differentiates itself from prior work with relevant citations.
- It uncovers Insightful findings, e.g., that model predictions often collapse to a few labels despite a large candidate set.
- It introduces the use of GradCAM to interpret which visual features influence the model’s decisions, adding novelty to the analysis.

Weaknesses:

- The analysis focuses solely on the image encoder, without exploring the multimodal layers responsible for aligning text and vision in VLMs. Applying interpretability techniques across the full model could offer a more comprehensive understanding of cross-modal associations.
- While the study is original, its significance is limited. It largely confirms existing suspicions about inconsistencies across models rather than investigating the reasons behind those. A deeper investigation into the causes of this misalignment (e.g., the surprising lack of symbolic alignment in Japanese) would have added more impactful insight.

Minor: on line 333, the claim that the ViT variant underperforms compared to ResNet is unclear. Figure 4 suggests that ViT yields more consistent results and performs better on English adjectives. Clarification is needed here.

---

> ### Author Rebuttal · Authors · 2025-07-28
>
> First of all, we want to thank you for taking the time to review our paper and share your opinion about it.
>
> # Weaknesses
> **Analyses on multi-modal layers –** More comprehensive investigation into intermediate layers is indeed an interesting avenue to explore. This would especially be interesting for dual-stream models that learn additional multi-modal layers between the text and vision modality (e.g. BLIP2) or single-stream models like VILT. However, the results of Verhoef et al. 2024 indicate that these models do not display behaviour consistent with a bouba-kiki effect. We, therefore, do not explore these models in this work and focus on CLIP.
>
> Since CLIP has been trained to map textual and visual inputs into a single joint multi-modal embedding, it does not have intermediate multi-modal layers that must explicitly align text and vision. In an attempt to better understand why we observe misalignment between human and CLIP’s behaviour, we explored whether the individual modalities are in principle separable in Appendix A & B and found that this seems to be the case. This indicates that CLIP’s method to align texts and images (cosine similarity) seems to be the culprit (please also see our response to your first question below).
> Nevertheless, your suggestion of investigating intermediate layers in dual-stream models is an exciting idea for future work.
>
>
> **Limited significance –** We argue that the significance is not limited. First of all, our Grad-CAM method is novel (which you identified as a strength of this paper) and goes beyond confirming inconsistencies by showing that CLIP does not attend to human-like image features (Figure 1 and Figure 10). Secondly, our work nuances the findings by Alper and Averbuch-Elor (2023), who make claims that require more substantiated argumentation. Our work offers contrary findings and accompanying argumentation that we see as a significant message for the community. Finally, we do address the surprising finding that Japanese VLMs do not exhibit expected bouba-kiki effects in lines 140-145. We draw on the findings by Iida and Funakura (2024) and conclude that investigating sound-symbolism in VLMs requires methodologies aligned with human experiments.
>
>
> **Clarification line 333 –** Thank you for this comment. Our phrasing should have been more concise here. The ViT variant indeed performs better on English adjectives, but *crucially* does not do so for the targets of interest: the pseudowords. The English adjectives merely serve as a baseline (line 187) to inform us whether the models can, in principle, recognise the relevant visual features and map them to meaningful words (which they can). The claim in line 333 corresponds to the performance on the pseudowords; we will make this clear in the revised version of this paper.
>
> # Questions
> **Reflection on misalignment –** This is a very interesting suggestion. We think that there are two reasons for the observed discrepancy. First, CLIP’s training data indeed does not explicitly reflect symbolic or phonetic associations. Nevertheless, human language and its orthography are not arbitrary either (see also our response to reviewer xGW7). Characters that represent rounded sounds (like rounded vowels or sonorant consonants) tend to have more curves (Koriat, 1977;Turoman & Styles, 2017). Since these sounds are related to shape features, CLIP’s training data could (but it doesn’t) still provide enough information to learn human-like cross-modal associations.
> Secondly, CLIP combines textual and visual information through cosine similarity and is trained with a contrastive loss at the sentence level. Neither seems to specifically promote learning relations between visual features and phonetic elements in language. Interestingly, however, as mentioned at line 247, the individual modalities seem to be separable in CLIP’s embedding space. We therefore expect that finetuning on (parts of) targeted datasets could reveal human-like associations. However, the bouba-kiki effect is merely one of many cross-modal associations present in human cognition. Instead of finetuning for each specific human-like bias, research towards a more fundamental approach that combines multi-modal information more naturally seems more desirable.
>
> **Tokenisation –** We want to thank you for your question and your argument that tokenisation oftentimes keeps syllabic structures intact. This increases its relevance and need for clarification.
>
> Interestingly, analyses reveal that humans even display a bouba-kiki effect at the level of single syllables (Nielsen and Rendall, 2013), suggesting that the associations do not rely on complete words. Since, as you mention, tokenisation often preserves syllabic structures, our analysis is not so different from investigating the bouba-kiki associations in humans.
>
> To add strength to this argument, we tokenised the most common labels mentioned in our answer to your first question. Some examples:
>
> Bouba → “bou” and “ba”
>
> Kiki → “kiki”
>
> Takete → “take”, “te”
>
> Maluma → “mal”, “uma”
>
> Xehaxe → “xe”, “ha”, “xe”
>
> Lohlah → “loh”, “lah”
>
> Sepise → “sep”, “ise”
>
> Kaykuh → “kay”, “ku”, “h”
>
> Loomoh → “loom”, “oh”
>
> Mohmah → “moh”, “mah”
>
> These examples, which we will add to an updated version of our paper, show that most of the target words are tokenised in a way that preserves at least some phonological structures that could be matched to shape features.
>
> Nonetheless, the tokenisation process in language models may split syllables or pseudowords into tokens that would not necessarily evoke the expected cross-modal associations in humans either (e.g., a separate evaluation of ‘H’ in OH may invite a jagged association instead of a curved one).
> If it is indeed the case that the tokenisation process breaks a pseudoword into syllables that would also show no bouba-kiki effects in humans, we would argue that the set of possible other pseudowords is large enough to contain other pseudowords that *could* invoke the cross-modal associations. If there would truly be a model preference, another pseudoword (which is tokenised into complete syllables) would then be preferred over the pseudoword that is broken up into non-syllables. Yet, our results show that this is not the case.
>
> To answer your question: While tokenisation may have an impact on the results, we think that the extent to which this influences them is somewhat limited. We did, therefore, not discuss this in detail in our paper, but we will do so as a response to your rightful question.
>
> # References
> Koriat, A. (1977). The symbolic implications of vowels and of their orthographic representations in two natural languages. Journal of Psycholinguistic Research, 6(2), 93–104.
>
> Nora Turoman and Suzy J Styles. 2017. Glyph guessing for ‘oo’and ‘ee’: Spatial frequency information in sound symbolic matching for ancient and unfamiliar scripts. Royal Society open science, 4(9):170882.

---

> > ### Comment · Reviewer_74nD · 2025-08-03
> > **Response to the authors**
> >
> > We thank the authors for their thorough responses and maintain our recommendation to accept the paper. The work presents an interesting research direction with insightful results.

---

> > > ### Author Response · Authors · 2025-08-06
> > >
> > > We are happy to read that your questions were resolved by our response. Thank you for your time and thoughts about our paper.

---

### Official Review · Reviewer_ZTfi · 2025-06-22

**Clarity:** 4
**Significance:** 3
**Originality:** 3
**Rating:** 5
**Confidence:** 4

**Summary:**

Authors test bouba-kiki effect, which is observed by humans, to vision language models. The authors throughly investigate the bouba-kiki effect on the vision language models, and conclude that the models do not present bouba-kiki effect, which implies that the models does not perceive the world as humans do.

**Questions:**

As I pointed out in the weakness, I would like to highlight two things:

1. Grad-CAM for transformer architecture and multimodal setting. I am curious if there is no issue of applying grad-cam to the transformers and multimodal models.
2. Why the current study is meaningful as the text-vision model is fundamentally different from human's world perception in the context of bouba-kiki effect? Especially when humans rely on audio instead of text, and when the audio plays a critical role in bouba-kiki effect, is it still meaningful to use text-vision model, not audio-vision model?

I am quite positive on this paper and its results, but if authors can make justifications on these points, I will gladly increase my score.

**Ethical Concerns:**

["NO or VERY MINOR ethics concerns only"]

**Final Justification:**

The response from the authors resolved my concerns. Specifically, the missing link between the written words to the sound, was pointed out by the authors, which made me understand the value of this work. The technical concern about the GRAD-CAM for transformer architecture is also answered. I believe this is a good work for Neurips.

**Limitations:**

Yes, authors listed limitations of the paper. But it would be further talk about the limitations as I described in the Question section. I believe those questions are crucial for this paper's validity.

**Paper Formatting Concerns:**

No concerns

**Quality:**

3

**Strengths And Weaknesses:**

## Overall
This is well written paper and readers would easily follow the logic. The topic of this paper covers an important topic that is often neglected in modern machine learning community. And the way this paper explores the topic is quite clear and through.

## Strength
1. Important topic and significance: Compared to the prior works, this paper conduct thorough study on bouba-kiki effect on the vision language models in more comprehensive and controlled ways. It challenges the belief that the vision language models work in a similar way to humans.
2. Rigorous methods: The authors employ a comprehensive set of pseudowords, multiple prompt templates, and image pairs inspired by human experiments, along with Bayesian regression analysis for statistically credible conclusions.
3. Clear writing: This paper is well written and easy to follow.

## Weakness
1. Grad-CAM for transformer architecture: The Grad-CAM method is originally proposed for convolutional operations. The use of grad-cam for the transformer architecture and multi-modal setting is not quite clear to me. If you could provide some justification on this, that would be helpful.
2. Connecting text and images, not speech: If my understanding is correct, the human studies listed in this paper are done by the speech, not text. Although humans are given texts, we can connect text to speech, and this speech in audio should deliver lots of precious information of whether the text or word is more like bouba or kiki. Although I think this work and the series of prior works are fascinating, I want to ask a fundamental question on this approach, that uses text instead of speech or audio. I understand that the typical models of multimodal processing are text and vision, not audio and vision. But is it truly fair comparison to human's world perception as we can connect text to audio, while vision-language models don't?

---

> ### Author Rebuttal · Authors · 2025-07-28
>
> First of all, we want to thank you for your feedback and are delighted to see that you are positive about our paper. Please find our justifications for your questions below.
>
> # Weaknesses
> **Grad-CAM for ViT –** It is indeed correct that Grad-CAM was originally introduced for CNNs. However, recent works (Chefer et al. 2021, Mamooler, 2021) have shown that it can be adapted to transformer-based models by treating attention maps as similar to convolutional feature maps. In our paper, following the approach in these previous works, we applied Grad-CAM to the last attention layer of the ViT variant of CLIP, which contains spatial information through its attention heads. We specifically focused on the attention from the [CLS] token to the image patches (like in Caron et al. (2021)). To identify which image regions contributed most to the decision, we computed the gradients of the similarity score between the image and text embeddings (which we used as the target for classification) with respect to the attention weights. Then, we averaged the attention maps across heads, extracted the attention weights from the [CLS] token to the image patches by removing the [CLS] column, and reshaped the resulting vector into a 2D spatial grid. We hope that this justification is satisfactory.
>
> **Audio-Vision methodology –**
> This is a very legitimate question. We agree with you that sound-vision entanglement is underexplored in our work and that it is a very interesting future direction. There are, however, some points that should be noted. While it is widely assumed that the bouba-kiki effect in humans is largely driven by sound-shape correspondences, many prior experiments with humans actually provide both an acoustic and written form for tested pseudowords (which was also the case in the study by Nielsen and Rendall (2013) which formed the basis for one of our word sets), so many prior studies cannot clearly disentangle the roles of orthography versus sound in the bouba-kiki effect. One study by Cuskley et al. (2017) explicitly explored the role of orthography and showed that auditory presentation cannot eliminate potential effects of orthography in literate human participants, since characters are typically strongly associated with a particular speech sound. Moreover, these associations are often not arbitrary but also shaped by human word-shape biases, resulting in systems where characters that represent rounded sounds (like rounded vowels or sonorant consonants) tend to have more curves (Koriat, 1977;Turoman & Styles, 2017). Correspondences between vision and spoken words are therefore expected to be highly correlated with correspondences between vision and written form. This makes it unlikely that a model trained on spoken language would be more likely to exhibit the bouba-kiki effect than a model trained on text would.
> Given the strong relationship between sound and wordforms and the fact that these words cannot be reliably associated with images, it would be surprising if an audio-visual model would pick this up. Despite our scepticism, we definitely think that this is worth exploring further with models that differ in how they combine multiple modalities.
>
> All in all, we believe that we can make a fair comparison. We will elaborate further on the dichotomy between human sound-shape associations and the current study, focusing on text-shape associations, in the limitations section of our updated paper.
>
> # Questions
> We believe that your questions are addressed in the response above, which addresses the weaknesses you identified. We are happy to answer any additional questions that may arise.
>
> # References
> Chefer, H., Gur, S., & Wolf, L. (2021). Generic attention-model explainability for interpreting bi-modal and encoder-decoder transformers. In Proceedings of the IEEE/CVF international conference on computer vision (pp. 397-406).
>
> Mamooler, S. (2021). CLIP explainability. Github repository
>
> Caron, M., Touvron, H., Misra, I., Jégou, H., Mairal, J., Bojanowski, P., & Joulin, A. (2021). Emerging properties in self-supervised vision transformers. In Proceedings of the IEEE/CVF international conference on computer vision (pp. 9650-9660).
>
> Koriat, A. (1977). The symbolic implications of vowels and of their orthographic representations in two natural languages. Journal of Psycholinguistic Research, 6(2), 93–104.
>
> Nora Turoman and Suzy J Styles. 2017. Glyph guessing for ‘oo’and ‘ee’: Spatial frequency information in sound symbolic matching for ancient and unfamiliar scripts. Royal Society open science, 4(9):170882.

---

> > ### Comment · Reviewer_ZTfi · 2025-08-02
> >
> > The response from the reviewer is convincing. Especially, I see some questions from multiple reviewers about the language models as a proxy to a sound models. The authors pointed out that there is a relationship between the word shape and sound based on the prior research, and it is something I wanted to know. All of my concerns are resolved and I adjusted my score. I would like to read more similar works from Neurips.

---

> > > ### Author Response · Authors · 2025-08-06
> > >
> > > We are happy to read that our response was satisfactory and resolved your questions. Thank you for your time and thoughts about our paper.

---

### Official Review · Reviewer_xGW7 · 2025-07-02

**Clarity:** 4
**Significance:** 2
**Originality:** 2
**Rating:** 4
**Confidence:** 5

**Summary:**

This paper revisits the classic bouba-kiki effect within the context of vision-language models (VLMs), particularly two CLIP variants (ResNet and ViT). Unlike previous studies that rely on embedding similarity or generation-based probing, this paper introduces a methodology tightly aligned with human psycholinguistic experiments.

The findings show that, despite some sensitivity to semantic adjectives (e.g., "round", "spiky"), CLIP models do not consistently exhibit the bouba-kiki effect for pseudowords, failing both in behavior and visual attention alignment.

**Questions:**

- Did you explore whether the CLIP models consistently output the same label regardless of the image input (i.e., label collapse), and if so, how does this affect interpretability?
- Did you analyze the spatial distribution of attention beyond total intensity? Your Grad-CAM analysis uses summed activation across curved vs jagged halves, but this could overlook spatial patterns within each region. For example, did you consider using centroid-of-attention, entropy, or overlap with curvature-dense regions to better quantify alignment with shape features?
- How does tokenization affect the model's ability to form shape-word associations, particularly for pseudowords? Since CLIP and other Transformer-based models operate on subword or byte-pair encodings (e.g., "bouba" might be split into “bou” and “ba”), this fragmentation may distort the phonological structure that underlies the bouba-kiki effect in humans, where sound symbolism operates over holistic auditory forms. Did you analyze how the pseudowords are tokenized by the CLIP text encoder, and whether token splits align with phoneme boundaries or disrupt the intended phonetic profiles? Could this explain the models' failure to form shape-sound associations, especially compared to their robust performance on real adjectives?

**Ethical Concerns:**

["NO or VERY MINOR ethics concerns only"]

**Final Justification:**

My concerns are addressed indeed, but no more exciting results are added during rebuttal. I would keep my current ratings.

**Limitations:**

Yes

**Quality:**

4

**Strengths And Weaknesses:**

## Strengths

- This paper stands out not for discovering a new bouba–kiki effect, but for disproving its presence in CLIP with a much more rigorous and cognitively aligned methodology. It raises the bar for testing cross-modal iconicity in vision-language models. Prior work (e.g., Alper & Averbuch-Elor) used embedding similarities or generation-based tasks with less experimental control. This paper brings direct comparability to human data, especially from Ćwiek et al. (2022).
- Human-aligned evaluation protocol: The authors closely mirror experimental designs used in psycholinguistics (e.g., forced-choice tasks with minimal shape differences), allowing for principled comparison between human and model behavior.
- A strong plus to use Grad-CAM for cross-modal interpretability: By examining visual attention patterns during label-shape association, the paper moves beyond accuracy to probe the internal mechanisms of cross-modal processing in VLMs.

## Weaknesses

- Sound-text-vision entanglement is underexplored: Since the original bouba-kiki effect is rooted in auditory perception, the textual substitution (e.g., "bouba" as a string token) may be too weak a proxy. The paper acknowledges this but does not systematically explore text-to-speech models or phoneme-level grounding.
- Model scope is limited to CLIP: While CLIP is foundational, the paper does not evaluate whether more modern VLMs (e.g., BLIP-2, LLaVA) or large multimodal models (e.g., GPT-4o) show similar or different patterns, which weakens the generality of the conclusions.
- While the paper introduces novel methodological angles, the core question of whether VLMs exhibit the bouba-kiki effect has already been substantially explored in prior work, particularly by Alper & Averbuch-Elor (2023) and Verhoef et al. (2024). As such, the contribution may feel incremental, and the bouba-kiki phenomenon, though conceptually interesting, may be too narrow in scope to carry a full conference paper on its own.

---

> ### Author Rebuttal · Authors · 2025-07-28
>
> First of all, we wish to thank you for the positive feedback. It gives us joy to read that you found our work interesting.
>
> # Weaknesses
> **Sound-vision entanglement –** We agree with you that sound-vision entanglement is underexplored in our work and that it is a very interesting future direction. There are, however, some points that should be noted. While it is widely assumed that the bouba-kiki effect in humans is largely driven by sound-shape correspondences, many prior experiments with humans actually provide both an acoustic and written form for tested pseudowords (which was also the case in the study by Nielsen and Rendall (2013) which formed the basis for one of our word sets), so many prior studies cannot clearly disentangle the roles of orthography versus sound in the bouba-kiki effect. One study by Cuskley et al. (2017) explicitly explored the role of orthography and showed that auditory presentation cannot eliminate potential effects of orthography in literate human participants, since characters are typically strongly associated with a particular speech sound. Moreover, these associations are often not arbitrary but also shaped by human word-shape biases, resulting in systems where characters that represent rounded sounds (like rounded vowels or sonorant consonants) tend to have more curves (Koriat, 1977;Turoman & Styles, 2017). Correspondences between vision and spoken words are therefore expected to be highly correlated with correspondences between vision and written form. This makes it unlikely that a model trained on spoken language would be more likely to exhibit the bouba-kiki effect than a model trained on text would.
> Given the strong relationship between sound and wordforms and the fact that these words cannot be reliably associated with images, it would be surprising if an audio-visual model would pick this up. Despite our scepticism, we definitely think that this is worth exploring further with models that differ in how they combine multiple modalities.
>
> **Limited model scope –** We indeed only test two CLIP variants for two reasons. First, CLIP is a foundational model that is often used in more contemporary models (lines 176-178). If this model does not show human-like associations, it is difficult to imagine that further-tuned models do show human-like preferences. Second, Verhoef et al. (2024) tested CLIP, VILT, BLIP-2, and GPT-4o in a similar vein to the first experiment in our paper. Out of these models, only CLIP and GPT-4o showed limited evidence of a bouba-kiki effect. Given the fact that we extend the existing approaches with an interpretability-based methodology to unravel the attention preferences of VLMs, we focus only on CLIP here because GPT-4o does not provide the required openness for this method. It goes without saying that newer methods are released as we speak (e.g., the pointing behaviour of AI2’s Molmo—which also uses CLIP as a visual backbone), which require further investigation.
>
> **Incremental contribution –** Since the topic of our paper is relatively new in this community (introduced by Alper and Averbuch-Elor at NeurIPS 2023), we would argue that it is still maturing, and we believe our contributions are more than incremental. This is also corroborated by reviewer 74nD, who states that our work differentiates itself from prior work. Given the strengths you identified (points 2 & 3), it seems to us that you agree with this. Finally, our methodology is strongly aligned with humans and can inspire others to align their methods with human experiments as well.
>
>
> # Questions
> **Label collapse –** This is super interesting, and we did indeed explore this in Appendix C. Table 2 shows that both CLIP versions tend to pick some words more often than others. However, out of 34 trials (images), the average ratio of ±33% across ten prompts indicates that around 10 different pseudowords are picked across all the images. As such, it seems like there is a preference for some pseudowords but no such thing as label collapse.
>
> As a response to your question, we also looked into the most frequently chosen pseudowords and provide some examples. For ViT and Alper et al’s pseudowords, some of the labels that appear most are “manuma”, “kiteki”, “gologo”, “xehaxe”, “lunulu”, “mogumo”. For ResNet these were: “xekaxe”, “sepise”, “xesixe”, “xehexe”, “gologo”. In the case of ResNet and Nielsen et al. pseudowords: “lahloh”, “mohmah”, “kaykuh”, “mohmoh”, “puhkuh”. And for ViT: “teepee”, “lohlah”, loomoh”, “lohlah”, “looloh”, “loomah”. The updated version of our paper will include this extra information.
>
> **Spatial distribution of Grad-CAM –** Thank you for the suggestion. This is indeed something we explored but did not mention in the paper since we argue that the sum of attention aligns with the holistic view humans have (lines 272-274). Nonetheless, we also used different spatial distributions, such as the centroid of attention and the entropy method you mentioned.
>
> Comparing the predictions chosen by the centroid of attention with the sum of attention, we find that 86.8% of the predictions are the same. Given this large overlap, the results likely would not change if we used this alternative method.
>
> In the case of entropy as a source of prediction (where the image half with the lower entropy acts as the model’s preference), we find that only 50.6% overlaps with the predictions based on the sum of attention. Yet, this method results in a lower number of correctly predicted images, with only 50.7% correctly identified shapes as opposed to 55.4% for the method which uses the sum of attention. So if we were to use entropy-based predictions, the performance would be even worse than predictions based on the sum of attention.
>
> **Tokenisation –** This is an interesting point that requires further clarification since it is not immediately clear whether tokenisation hurts or benefits bouba-kiki-like associations (see also the question of reviewer 74nD).
>
> Interestingly, analyses reveal that humans even display a bouba-kiki effect at the level of single syllables (Nielsen and Rendall, 2013), suggesting that the associations do *not* rely on complete words. Since tokenisation often preserves syllabic structures, our analysis is not so different from investigating the bouba-kiki associations in humans.
>
> To add strength to this argument, we tokenised the most common labels mentioned in our answer to your first question. Some examples:
>
> Bouba → “bou” and “ba”
>
> Kiki → “kiki”
>
> Takete → “take”, “te”
>
> Maluma → “mal”, “uma”
>
> Xehaxe → “xe”, “ha”, “xe”
>
> Lohlah → “loh”, “lah”
>
> Sepise → “sep”, “ise”
>
> Kaykuh → “kay”, “ku”, “h”
>
> Loomoh → “loom”, “oh”
>
> Mohmah → “moh”, “mah”
>
>
> These examples, which we will add to an updated version of our paper, show that most of the target words are tokenised in a way that preserves at least some phonological structures that could be matched to shape features.
>
> Nonetheless, the tokenisation process in language models may split syllables or pseudowords into tokens that would not necessarily evoke the expected cross-modal associations in humans either (e.g., a separate evaluation of ‘H’ in OH may invite a jagged association instead of a curved one).
> If it is indeed the case that the tokenisation process breaks a pseudoword into syllables that would also show no bouba-kiki effects in humans, we would argue that the set of possible other pseudowords is large enough to contain other pseudowords that *could* invoke the cross-modal associations. If there would truly be a model preference, another pseudoword (which is tokenised into complete syllables) would then be preferred over the pseudoword that is broken up into non-syllables. Yet, our results show that this is not the case.
>
> # References
> Koriat, A. (1977). The symbolic implications of vowels and of their orthographic representations in two natural languages. Journal of Psycholinguistic Research, 6(2), 93–104.
>
> Nora Turoman and Suzy J Styles. 2017. Glyph guessing for ‘oo’and ‘ee’: Spatial frequency information in
> sound symbolic matching for ancient and unfamiliar scripts. Royal Society open science, 4(9):170882.

---

### Official Review · Reviewer_BFRt · 2025-07-02

**Clarity:** 3
**Significance:** 3
**Originality:** 3
**Rating:** 4
**Confidence:** 4

**Summary:**

The paper investigates whether vision-and-language models (VLMs), specifically two variants of CLIP (ResNet and Vision Transformer), exhibit human-like cross-modal understanding by testing the bouba-kiki effect—a phenomenon where people associate round shapes with the word "bouba" and jagged shapes with "kiki." Using prompt-based evaluation and Grad-CAM for visual attention, the study finds that:
* ResNet shows some preference for round shapes, but
* The models do not consistently demonstrate the bouba-kiki effect.
* Their performance falls short of human-like cross-modal integration.
These findings suggest that current VLMs have limited internal representations of such intuitive, cross-modal associations, raising questions about the depth of their "understanding."

**Questions:**

- Please motivate why the work is valuable for the machine learning community as you do not propose strategies to improve cross-modal associations.
- What is the practical impact of investigating the bouba-kiki effect?

**Ethical Concerns:**

["NO or VERY MINOR ethics concerns only"]

**Final Justification:**

The answers to my questions mitigate my concerns.

**Limitations:**

The limitations of the work are discussed.

**Paper Formatting Concerns:**

There are no formatting issues.

**Quality:**

3

**Strengths And Weaknesses:**

Strengths:
- The paper is well written and easy to follow.
- The paper provides a good overview of the literature regarding the bouba-kiki effect.
- The results are obtained both with synthetic and natural data.

Weaknesses:
- The paper does not contribute to novel machine learning models. Its contribution is interesting for the natural language processing and computer vision domains, so EMNLP, CoNLL and CVPR might be better venues for its publication.
- It is strange that a CLIP model was chosen as underlying vision-language foundation model. It is well known that alignment in CLIP is computed at the sentence/scene level and not at the level of individual words, which devalues the findings of the paper.

---

> ### Author Rebuttal · Authors · 2025-07-28
>
> First of all, we would like to thank you for the feedback. We are happy that you had only a few weaknesses/questions, that the paper was easy to follow, and that you enjoyed the overview of the literature. Please find our response to the weaknesses and questions below.
>
> # Weaknesses
> **Contribution to machine learning models –** We acknowledge that our work is primarily in the domain of vision and natural language. Nevertheless, there are several reasons why we think it is as interesting and relevant for the NeurIPS community as for venues like EMNLP or ACL. Our work fits squarely within several themes listed in the CfP, such as ‘deep learning’, ‘neuroscience and cognitive science’.  Moreover,  the NeurIPS community oftentimes advances deep learning techniques (generative AI, foundation models, LLMs, etc.). Our work reveals an important shortcoming in current approaches, which will hinder the continued development of models that aim to align human and computer visio-linguistic, and more generally multimodal, processing. This is arguably equally important as introducing new methods, as evidenced by the fact that Alper and Averbuch-Elor (2023)—whose conclusions appeared to have been drawn somewhat prematurely, given our new findings—also published their findings (spotlight) at NeurIPS.
> Finally, as also pointed out as a strength by reviewer xGW7, our work evaluates the same problem via different methods to provide a more comprehensive and nuanced picture. Such practices are, in our opinion, of value to the NeurIPS community as we believe that our findings not only could inspire others to develop models that take cross-modal associations into account, but also introduce several ways to test for such associations in a human-aligned manner.
>
> **CLIP as tested model –** There seems to be a misunderstanding. You are completely correct that alignment in CLIP is calculated at the sentence level. As spelled out in lines 202-204 and shown in table 1 (appendix A), we indeed embed the pseudowords into sentences (10 versions) such that this alignment score can be calculated at the sentence level. Importantly, for each image, we *only* differ the pseudoword in question, so variation in probability *must* be a result of the pseudoword. We hope that this alleviates your concerns about the use of CLIP.
>
> # Questions:
> **Contribution and value of this work –** Please see our answer as a response to weakness 1.
>
> **Practical impact –** The importance of shared perception/experiences/assumptions is often overlooked in the machine learning community (see also reviewer ZTfi). The bouba-kiki effect, or cross-modal associations in general, is one example for which we would like there to be alignment between humans and models since this results in a more intuitive understanding of each other’s prior expectations and assumptions. A lack thereof may hinder natural interactions between humans and machines (Verhoef et al. 2024), especially when considering the dynamic and adaptive nature of human language. Training models in a way that allows them to develop human-like prior expectations or instilling models with them may help alleviate this and could even improve learning efficiency (Lake et al., 2017). The practical impact of our work will be incorporated in the updated version of our paper.
>
> # References
> Lake BM, Ullman TD, Tenenbaum JB, Gershman SJ. Building machines that learn and think like people. Behavioral and Brain Sciences. 2017;40:e253. doi:10.1017/S0140525X16001837

---

> > ### Author Response · Authors · 2025-08-06
> >
> > Dear reviewer,
> >
> > We would like to hear whether our response to your concerns was satisfactory or whether you have further questions. We are happy to engage in further discussion.

---

> > ### Comment · Reviewer_BFRt · 2025-08-06
> >
> > I am happy with the responses.

---

### Note · Authors · 2025-08-13

Dear reviewers, AC and SAC,

We are pleased with the reviews and the rebuttal period. Though we did not always have a lengthy discussion, the rebuttals resolved the concerns of engaged reviewers (e.g., "I would like to read more similar works from Neurips", "All of my concerns are resolved", "The response from the reviewer is convincing", "We thank the authors for their thorough responses and maintain our recommendation to accept the paper", and "I am happy with the responses"). The lack of active discussion for other reviewers seems to *imply* that our rebuttal also resolved all of their initial concerns.

Overall, we are happy with the questions raised. We believe that incorporating our responses in an updated version of the paper truly benefits its quality and clarity, for which we are thankful.


Best regards,
The authors

---

### Decision · Program_Chairs · 2025-09-17

**Decision:**

Accept (poster)

**Comment:**

(a) **Summary:** This paper explores the extent to which the bouba-kiki effect – where humans associate “bouba” with round shapes and “kiki” with jagged shapes – is present in CLIP models. They use both model probabilities and GradCAM to test model preference and visual attention, respectively. They found CLIP with the ResNet backbone has some preference for round shapes but overall, the models do not demonstrate the bouba-kiki effect. Importantly, CLIP models do not show the same cross-model behavior for this effect as humans do. It suggests models do not demonstrate the same human intuitions.

(b) **Strengths:**
- This work uses both synthetic and natural data (**Reviewer BFRt**), showing the findings are generalizable to different types of data.
- The methodology for the models is a forced choice task, which closely aligns with existing psycholinguistics literature (**Reviewer xGW7**).
- By using GradCAM, this paper explores more of the interpretability angle that is not possible from model probabilities alone (**Reviewer xGW7, Reviewer 74nD**).
- The paper is well-written overall and easy for readers to understand (**Reviewer BFRt, Reviewer ZTfi**).

(c) **Weaknesses:**
- Only CLIP models are used (**Reviewer BFRt**), while more SOTA VLMs are excluded from the analysis. I agree that additional VLMs should be tested, especially given the title and wording in the abstract.
- The findings are a negative result in some ways (CLIP models do not show the bouba-kiki effect) without giving insight into why it happens and what the significance of this is (**Reviewer 74nD**). The paper would benefit from a deeper analysis about why the misalignment with humans happens.

(d) **Recommendation:** Reviewers overall view this work positively, with scores of 4, 4, 5, 4. The strengths are in the experimental setup, where both probabilities and attention are used to probe the bouba-kiki effect. Additionally, the work thoughtfully parallels existing human studies. The weaknesses are that only CLIP models are used. While I appreciate that the authors used two backbones, one CNN-based and one transformer, I do agree that more VLMs should have been used. This would strengthen the conclusions and give more insight into other widely used, SOTA models. With all of this considered, this work has solid findings supported by the reviewers. I suggest acceptance.

(e) **Discussion/Changes during Rebuttal:**
- **Point:** GradCAM was originally proposed for convolutional operations and therefore may not be transferable to the ViT backbone. In the rebuttal, the authors present additional work that uses GradCAM for Transformer models. The reviewer suggests they are convinced by the rebuttal. Impact: This was fully addressed and therefore did not impact my suggestion.
- **Point:** CLIP embeds image-text similarity at the sentence level, so **Reviewer BFRt** is concerned about how the individual words’ impacts are tested. The authors clarify that only an individual word is toggled between sentences, such that the impact must be from said word. Impact: This clarification makes sense and this concern did not impact my recommendation.